# Motivation and Engagement of Final-Year Students When Using E-learning: A Qualitative Study of Gamification in Pandemic Situation

**Flourensia Sapty Rahayu** [1,2,*], **Lukito Edi Nugroho** [1], **Ridi Ferdiana** [1] **and Djoko Budiyanto Setyohadi** [2]

1    Departemen Teknik Elektro dan Teknologi Informasi, Universitas Gadjah Mada, Yogyakarta 55281, Indonesia; lukito@ugm.ac.id (L.E.N.); ridi@ugm.ac.id (R.F.)
2    Departemen Informatika, Universitas Atma Jaya Yogyakarta, Yogyakarta 55281, Indonesia; djoko.budiyanto@uajy.ac.id
*    Correspondence: sapty.rahayu@uajy.ac.id

**Abstract:** The COVID-19 pandemic has changed how the education system operates. The shift from face-to-face learning to online learning generated many problems, including decreasing students' motivation and engagement. Gamification has been used as one of the solutions to overcome the problem of low motivation and engagement in learning. The current study aims to examine students' behavioral change when using e-learning with gamification, investigate gamification elements that are important to students and how it influences students' motivation and engagement, and investigate whether population characteristics may influence students' motivation and engagement. Qualitative methods were employed to gather and analyze the data. The thematic analysis resulted in six main themes. The findings revealed that there were behavioral changes in students during gamification implementation, i.e., from negative to positive and from positive to negative. Four gamification elements were found to be the most important gamification elements to students, i.e., points, leaderboard, badges, and gamified test. The mechanism of how these elements influenced motivation and engagement was discussed. The population characteristics of final-year students also had an impact on gamification effectiveness. Despite gamification's capabilities to influence motivation and engagement, there are some concerns related to negative impacts that must be addressed in the future.

**Keywords:** gamification; e-learning; motivation and engagement; final-year students; the COVID-19 pandemic

## 1. Introduction

The COVID-19 pandemic has disrupted education systems globally, which caused educational institutions' closure [1]. More than 87% of the world's student population, over 1.5 billion learners in 165 countries, have been affected by the temporary closure of educational institutions [2]. Education systems were forced to shift from face-to-face learning to remote learning. The rapid changes forced teachers and students to adapt to the new situation abruptly [3], which has generated new problems. The Association for Psychological Sciences has summarized some impacts of the pandemic on both children's and adults' mental health [4]. These impacts include loneliness, increased levels of stress, anxiety, and depression. A review study on teaching and learning during the COVID-19 pandemic found that students' anxiety and stress were triggered both by technical limitations, as well as concerns about assessments and the achievement of learning goals [5].

Students' continuous stress may affect both academic performance and mental and physical health [6]. A previous study suggested that students' motivation needs to be encouraged to reduce frustration and boost self-regulation and flexibility [4]. On the other side, the sudden shift to online learning greatly influenced learning motivation, which

could influence learning achievements, learner satisfaction, and learner participation in online learning environments [7]. A previous study in India found that the majority of the respondents said they had low motivation to study due to a lot of distractions at home [8]. A similar study found that children were significantly less motivated to learn during the pandemic [9]. A study of Norwegian students' experiences of homeschooling during the pandemic revealed that all student groups prefer regular school over homeschooling, and it was even harder for low-achieving students to maintain engagement and motivation during homeschooling compared to regular school [10]. Another study involving 539 college students in Indonesia found that students' motivation gradually decreased until one year after the initial use of online learning [11]. Sofianidis et al. [12] found some students' concerns about online learning during the pandemic. The first concern is that students considered online learning as not as effective as face-to-face instruction. Instruction in distance education was less satisfactory, less meaningful, and less structured than face-to-face instruction. The instructor might find it difficult to detect non-verbal cues regarding a student's attention, which made the instructor unable to determine whether the student is paying attention or not [13]. The other concern is related to the lack of socialization during online learning. Due to the many challenges students face with online learning, students are expected to be more responsible for their learning strategies, learning-related emotions, self-regulation, and motivation [13]. On the other side, educational institutions have tried to develop various strategies to ensure active student engagement in online learning, including developing a framework that combines the balanced use of adjusted teaching pedagogy, educational technologies, and an e-learning management system [14].

Since its introduction, gamification is still gaining researchers' attention due to its potential to increase user engagement and motivation in various domains [15], including in education [16,17]. Gamification refers to the use of game elements that are applied to non-game contexts [18]. The trend of gamification use during the pandemic was increasing [7,19]. However, previous studies that investigated the influence of gamification on motivation and engagement revealed various findings [18,20,21]. Seaborn and Fels [18] conducted a review study on gamification in 31 studies and found various results regarding the relationship between gamification and engagement. Several studies on gamification showed positive results, whereby gamification succeeded in increasing students' motivation to learn [22–24]. Groening and Binnewies [24] found that achievements in a gamified system could improve user performance and motivation if designed properly. Sousa-Vieira et al. [25] developed a software platform that combines a learning management system (LMS), an online social network, and gamification elements. The results of the three years of implementation showed that the system could increase student motivation and improve student learning experience and performance. Similar results were found by Jurgelaitis et al. [26] who showed that the use of gamification could improve student performance and intrinsic motivation. However, not all gamification implementations showed positive results. Hanus and Fox's research [27] found that students in gamification classes showed decreased motivation, satisfaction, and enthusiasm over time compared to students in classes without gamification. Research by Kyewski and Krämer [28] found that the use of badges had no impact on student's motivation and performance. Furthermore, research shows that students in the non-gamified group tend to be more active than students in the gamified group. Several studies have shown that the results of gamification may not be long term, but only in the short term which may result from the novelty effect [29,30]. Even the trend of using gamification was criticized because it was considered to only add game elements in existing applications as a cosmetic aspects, or to exclusively exploit behavioral principles to force user performance [31].

The selection of gamification elements plays a role in determining the effectiveness of gamification. A review study by Saleem et al. [32] stated that the selection of the right gamification elements will support and motivate students to participate in the gamification system. On the other hand, inappropriate gamification elements will not affect students' motivation or may harm students' motivation or performance. Previous studies that used

limited elements of gamification or that forced students to use available game elements found negative or mixed results (positive and negative) [33,34]. Previous research suggests that future gamification studies should investigate the elements of gamification more specifically rather than viewing it only as a comprehensive concept so that the effectiveness of each element of gamification can be known [27,35]. Moreover, Alsawaier [20] stated that there is a need to find the most effective gamification elements to provide conditions that allow for increasing intrinsic motivation.

The use of gamification is expected to provide an enjoyable experience in learning. Monotonous and repetitive activities that lack complexity, variety, and cognitive stimulation are identified as causing boredom [36]. Boredom and apathy are considered the reasons many students are not interested in learning [37]. The elements of gamification may turn boring activities in learning into interesting activities [38], thus bringing enjoyable experiences. However, gamification still raises skepticism about its effectiveness and ability to provide a truly enjoyable experience for users [31]. Rapp [31] stated that one of the shortcomings of gamification techniques is the limited types of game elements available for game designers to use [39], combined with a lack of understanding of how these elements can impact the user's subjective experience. Furthermore, Laschke and Hassenzahl [40] stated that the goal of gamification should not only be to show and maximize the change of a certain behavior, but must also make the change a valuable experience. Gamification itself aims to make users have a pleasant experience, so to replicate it in a non-game context, developers must first understand how this experience can be formed from within [31]. This experience may be derived from another domain that has been proven to successfully engage users, i.e., the video games domain. Thus, the game elements used in video games that have been proven to influence users' engagement may be replicated in a non-game context.

We have developed a gamification system using gamification elements adapted from video game elements that have been proven to influence engagement. The experiment of using the gamification system was conducted in a research methodology class, Information System Program, Atma Jaya Yogyakarta University, Indonesia, in the odd semester of 2021/2022. During that time, the learning process was still held fully online due to the pandemic. We have conducted quantitative and qualitative studies to investigate the effectiveness of gamification in influencing students' motivation and engagement. In this paper, we focus on the qualitative analysis of gamification by exploring the subjective experiences of users when using a gamification system. We argue that the users' subjective experiences, as well as the users' meanings, perceptions, and feelings, are significant in discovering factors that influence motivation and engagement in current gamified systems. However, other factors may also influence gamification effectiveness. Alsawaier [20] suggested that contextual differences may cause variations in the degrees to which a successful gamified experience is created. The contextual differences may include implementation differences, instructor's characteristics, and student's characteristics, i.e., readiness and willingness characteristics, prior experiences and exposure to video game elements, and willingness to engage. Another study found that some influencing characteristics include player type, age, gender, motivation, personality, and culture [41]. Besides individual characteristics, population characteristics may also be the influencing factors of gamification success. Urh et al. [42], in their study seeking to design a model of e-learning with gamification involving college students, stated that students in higher education are more aware of the importance of the education they have chosen. They have formed personal goals and career orientation, thus gamification must be designed to reinforce students' feelings of the importance of education for the future. Students in higher education itself may also have different characteristics; for instance, final-year students have more awareness of the importance of their career orientation than first-year students. The current study involved final-year students as participants; therefore, we expect that the characteristics of final-year students may impact the gamification results. Thus, the aims of this study are to (1) examine the students' behavioral change when using e-learning with gamification, (2) investigate

gamification elements that are important to students and how they influence students' motivation and engagement, and (3) investigate whether population characteristics may influence students' motivation and engagement.

## 2. Materials and Methods

The present study is a part of a study to implement a gamification system in e-learning to increase students' motivation and engagement, which has been performed via an experimental approach. As many as 22 students who attended a research methodology class participated in the experiment. According to Cohen et al. [43], the experimental methodologies require at least 15 participants, thus the number of participants was considered adequate to conduct the experiment. The outcomes of the experiment may be influenced by a variety of factors. Previous research involving students from Atma Jaya Yogyakarta University, where the present study population comes from, revealed that social factors highly influenced students' intention and behavior to use e-learning [44]. Due to the characteristics of the population, we limited the number of participants to minimize the influence of social factors on the expected results.

In the present study, the results of the experiments were validated using a qualitative method. Qualitative research focuses on describing and understanding the meanings people attach to their encounters with other people, their cultural environment, and material objects [45]. Qualitative methods, such as in-depth interviews and focus group discussions, are used to answer questions about experience, meaning, and perspective, most often from the participant's standpoint [46]. The qualitative approach provides a rich, contextualized understanding of some aspects of human experience as derived from particular cases. Since we investigated the students' experiences in a particular case, the qualitative method was considered suitable to be used. In particular, this study used reflexive thematic analysis (RTA) to perform data analysis. RTA is a qualitative approach that highlights the researcher's active role in knowledge production [47]. RTA fully embraces qualitative research values and the subjective skills the researcher brings to the process [48]. Codes in RTA are a representation of the researcher's interpretations of meaning across the dataset. Themes were the outcome of data coding and iterative theme development [48]. In RTA, the size or frequency is not the only (or even primary) determinant of theme development [49]. The coding quality of RTA did not stem from the sample size, but rather from the depth of the researcher's engagement with the data and situated, reflexive interpretation [50]. Since in RTA, meaning is generated through the "interpretation" of data, and is not "excavated" from data, the judgments about how many data items to use and when to stop data collection are inescapably situated and subjective [50].

The research methodology class was held during the odd semester of 2021/2022. During that period, all the learning activities in the university were still held fully online due to the pandemic. The final-year students here were students who were in their fourth year of study. The research methodology focused on a 7th-semester course that students must take as a mandatory course before undertaking their theses. Some students take the research methodology class along with an internship, while some other students only take this course in that semester. The current study employed qualitative methods, i.e., Focus Group Discussion (FGD), semi-structured interviews, and observation, to collect primary data. The first FGD was attended by 22 students, while the second FGD was attended by 10 students. The selection of 10 students who would attend the second FGD used judgmental sampling where the first author made a selection based on her professional judgment. Students selected for the second FGD and interviews were those who represented three groups of point-earning, i.e., the low, medium, and high point-earning groups in the gamification system. Interviews were held to obtain more in-depth information from the 10 informants' experiences. The FGDs and the interviews we held online via a Microsoft Teams meeting. The 10 informants were composed of 80% males and 20% females. As complementary data, we used the students' comments from the e-learning

forum. Researchers also employed participant observation to observe student behavior during gamification implementation.

Primary data from the FGDs, interviews, and forums were then analyzed using reflexive thematic analysis (RTA). Reflexive thematic analysis is an interpretivist paradigm, which emphasizes the understanding of individuals' subjective experiences. Since the authors were investigating the "meanings" that participants created and attributed to their experience in using the gamification system, RTA was considered an appropriate analytical approach. Braun and Clarke [48] proposed 6 phases for reflexive thematic analysis, namely, (1) data familiarization and writing familiarization notes; (2) systematic data coding; (3) generating initial themes from coded and collated data; (4) developing and reviewing themes; (5) refining, defining and naming themes; and (6) writing the report. Before processing the data, the interviews and FGDs recordings were transcribed. In the first stage of the thematic analysis, the first author read through all the transcripts twice to become familiar with the data. The next stage was data processing using NVivo software. Three phases are involved in data processing with NVivo, namely, (1) generating codes from the data, (2) reviewing and organizing the codes, and (3) developing themes. In the first phase, the first author assigned a code for each data point. Coding in RTA is open and organic, with no use of any coding framework. The coding generation used semantic coding and latent coding. Semantic codes are identified through the explicit or surface meanings of the data, whether latent codes are created by identifying hidden meanings or underlying assumptions or ideas that may shape or inform the descriptive or semantic content of the data. The initial codes were then reviewed and organized by merging, deleting, and recoding the codes. The next phase was the development of the themes. The codes were collected under the developing themes. Themes in RTA are patterns of shared meaning, united by a central concept or idea [48]. We developed themes from coded data that have shared meanings. This process involved collapsing multiple codes that share a similar underlying concept or feature of the data into one single code [47]. The initial themes were then reviewed and refined by other authors. For writing the report, we used direct quotes from the respondents to illustrate the various topics that arose during the FGDs and interviews. Each quote has been assigned a coding reference that refers to the respondent's number (i.e., R2, refers to respondent 2).

### 3. Gamification Implementation in E-Learning

The gamification was implemented in Moodle-based e-learning. The Level Up!Plus plugin was installed in Moodle to accommodate the gamification elements. The selection of gamification elements to be implemented was based on previous studies on the video game domain (Table 1), which have identified particular video game features and game practices that are considered to influence video game engagement or problematic video game playing [51–54]

**Table 1.** Previous Studies on Video Game Elements that Influence Engagement.

| Study | Methods | Video Game Elements that Influence Engagement | Number of Participants | Participants' Age |
|---|---|---|---|---|
| King et al. [51] | Quantitative | Reward | 421 | Mean = 22.8, SD = 5.6 |
| Hull et al. [52] | Quantitative | Social | 110 | Mean = 24.7, SD = 9.04 |
| Laffan et al. [53] | Quantitative | Presentation and Punishment | 207 | Mean = 25.47, SD = 5.6 |
| Rapp [54] | Qualitative | Empowering, Farming, and Raiding | 36 | Mean = 28.3 |

Video game features or structural characteristics of video games are those features inherent within the video game itself that may facilitate the initiation, development, and maintenance of video game playing over time [55]. Game practices are the practices (activities) that players undertake in the game [54]. Learning from video games, our study

adapted those video game features and game practices in an e-learning context to see whether the implementation of those features and practices could help increase students' motivation and engagement as it did in video games.

We used the taxonomy of video game structural characteristics [55] to classify gamification features. According to the taxonomy, video game structural characteristics include reward and punishment, social, presentation, narrative and identity, and manipulation and control features. Reward and punishment features reward players for skillful play and punish players for losing. Social features are the socializing aspects of video games. Presentation features provide the aesthetic qualities of a video game. Narrative and identity features allow a player to can take on another identity in the game. Manipulation and control features are related to how a player can interact with and control in-game properties using a physical control scheme.

Based on previous studies, we selected the reward features [51], social features [52], and three-game practices, i.e. empowering, farming, and raiding [54] to be implemented in the gamification system. The presentation features were not selected to be implemented in e-learning due to the difficulty of implementation in Moodle-based LMS. The punishment features were also not implemented since they may bring negative consequences to learning [56]. We grouped the video game structural characteristics groups reward features into general reward, meta-game reward, intermittent reward, and payout interval. The gamification elements selected and their implementation are shown in Table 2.

**Table 2.** Gamification Elements and their Implementation.

| Gamification Element | Element Type | Sub-Elements | Implementation |
|---|---|---|---|
| Social | Feature | Social utility | Online chat |
| | | Social formation and institutional | Team |
| | | Leaderboard | Individual leaderboard, team leaderboard |
| | | Support network | Forum |
| Reward | Feature | General reward | Experience Point (XP) |
| | | Meta-game reward | Badges, bonus points, social point |
| | | Intermittent reward | Reward schedule (periodic and variable) |
| | | Payout interval | Lag time for giving rewards |
| Empowering | Practice | | Gamified test, assignment |
| Farming | Practice | | Wiki, display/download materials |
| Raiding | Practice | | Team discussion, team battle |

The general reward was implemented by experience points (XP), which were given regularly. Students received some points according to what activities they performed. The points were given after the activity was completed. For instance, 20 points were given after students finished their post-test every week. The meta-reward was represented by badges, levels, social points, and bonus points. Seven distinctive badges were provided for the students as a tribute to their achievement. The seven badges and their requirements are shown in Table 3. The badges can be classified into three groups: (1) competency badges, (2) social badges, and (3) participation badges. Competency badges were given to students who successfully reached a certain level or leaderboard position. For instance, champion badges were given to students that successfully maintained a position in rank 1–5 for 7 consecutive weeks. Competency badges included champion, team champion, and top scorer badges. Social badges are badges that are given to students for their social behavior. For instance, generosity badges were given to students for their generosity in giving points to their friends. Participation badges were rewarded for students' participation in the class. These badges included persistence, speed, and activeness badges. The level represents

a particular difficulty level. The difficulty level was implemented by higher points to be achieved in the next level. Reaching the next level is a reward for students' achievement. In the e-learning gamification implemented, there were 10 levels, and students had to achieve level 5 as a minimum requirement to pass the course.

**Table 3.** Badges List.

| Badges Icon | Badges Name | Description |
|:---:|:---:|:---:|
|  | Champion | Award for successfully maintaining an individual's position in rank 1–5 for 7 consecutive weeks |
|  | Team Champion | Award for successfully maintaining the team's position in rank 1–3 for 7 consecutive weeks |
|  | Top Scorer | Awards for the first 3 students who managed to reach level 10 |
|  | Persistence | An appreciation for the diligence and persistence of students in accessing activities on the e-learning site |
|  | Speedy | Will be given to students who input/update the Wiki no later than Friday, before 24.00 every week for 4 consecutive weeks |
|  | Generosity | Will be given to students for their generosity in giving points to their friends |
|  | Activeness | Award for student activeness. Activeness can be seen from the number of chats and posts in the forum |

The social points were related to students' positive social behavior. For instance, an extra point was given to students who gave congratulations to their classmates for their classmate's achievements. The bonus point was given for certain students' achievements. For instance, if students got 100 in their pre or post-test, they received an additional 20 points. The intermittent reward was the schedule arrangement for rewarding. The payout interval was implemented by giving points right after students completed an activity, except for the chat activity. The point for chat activity was given manually since the additional Level Up!Plus feature could not detect the Moodle build-in chat feature as an activity that could be set up for generating points. Social features were implemented by several mechanisms. The social utility feature that accommodates social interaction with others was implemented by the chat feature. Students could use the chat to interact with the lecturer and other students. The social formation and institutional features were represented by students' teams. Each student joined a team that consisted of 2–3 students. This team was created to accommodate students' needs to belong to a social group. The implementation of leaderboard features included individual leaderboards and team leaderboards. Students could see their achievement rank every week on the leaderboard. The support network feature was implemented by Forum. In this forum, students could discuss things related to the class and give support to each other.

Besides game features, three game practices were implemented in the gamification system, i.e., empowering, farming, and raiding. The empowering practice aimed to increase students' competency. Some activities to empower the students included reading the material and doing tests and assignments. Instead of regular quizzes, we used various

gamified quizzes every week for students' tests. Students received some points after doing the activities. The bonus points were given to students that gained the highest score on tests. The farming practice was implemented by employing repetitive activities to collect points, for instance, displaying the material and posting wikis regularly. Since these activities can be done repetitively, the system set a small point, e.g., 5 points for these activities. The raiding practice aimed to foster students' collaboration with their teammates. The activities related to raiding were team discussions and team battles. Team discussions were held five times in one semester. Students were given particular topics to be discussed in their team. Team battles were held two times, one before mid-semester and one before the end semester. In this activity, each group of students competed to win the Team Champion badge and some bonus points. The battles were carried out in the form of game-based quizzes.

Figure 1 shows the gamification model that we have implemented. Two blocks, namely, gamification elements and gamification goals, constructed the model. The gamification elements block consists of game features (reward and social) and game practices (empowering, farming, and raiding). The dotted line in Figure 1 represents the relationship between elements; for example, in the relationship between empowering practice and reward feature, students were given some points after they performed the gamified test. Likewise, a team (social element) was needed for the team discussion or team battle activity (raiding practice). The gamification elements altogether were used to achieve the gamification goals, i.e., to increase students' motivation and engagement.

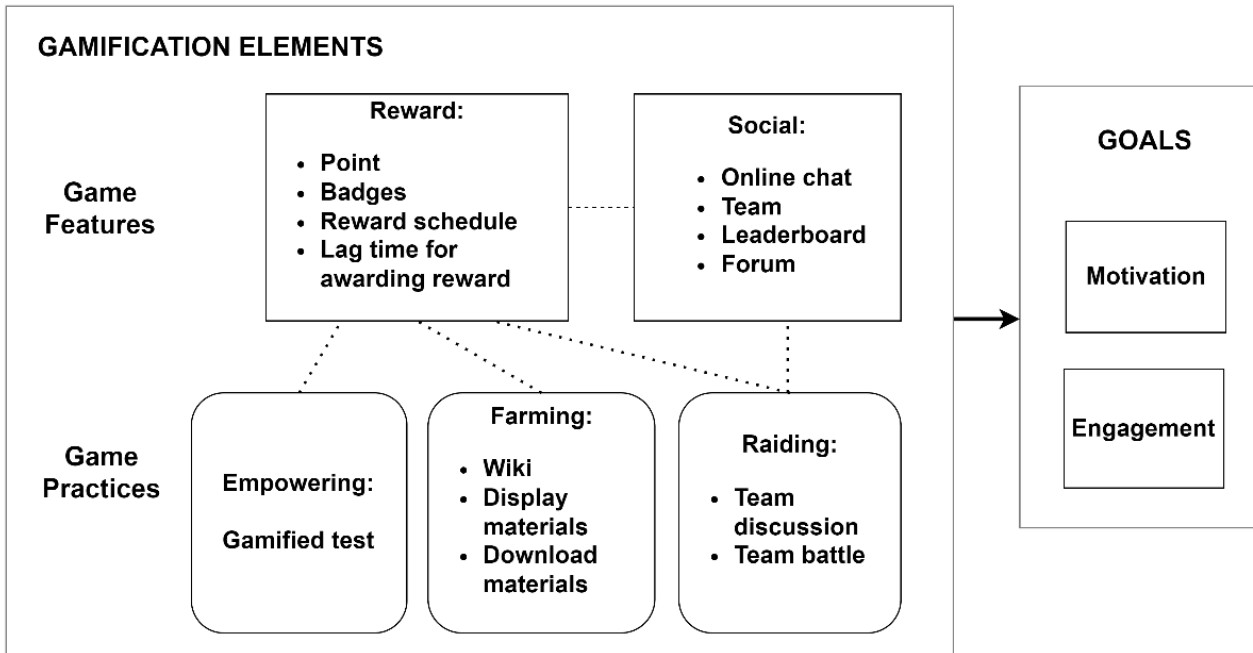

**Figure 1.** Gamification Model.

## 4. Results and Discussion

The goals of the present study were to (1) examine the students' behavioral change when using e-learning with gamification, (2) investigate gamification elements that are important to students and how they influence students' motivation and engagement, and (3) investigate whether population characteristics may influence students' motivation and engagement. Our thematic analysis generated six main themes that can be seen in Figure 2. In this section, we will present the results and discussion, structured according to the research goals.

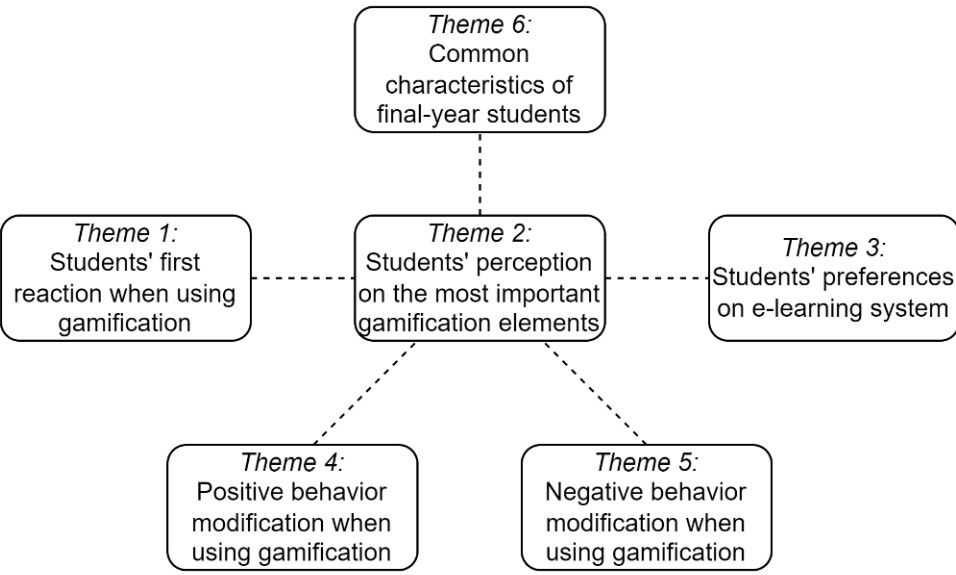

**Figure 2.** Thematic Framework.

*4.1. Behavioral Change*

In this sub-section, we will present the themes that are related to students' behavioral change, namely, theme 1, theme 3, theme 4, and theme 5. In theme 1, students express their reaction to their first introduction to the gamification system. Related to the first experience, all students agreed that this was their first experience using a gamification system for learning. Various reactions emerged when they used gamification e-learning for the first time; 20% of the participants experienced positive reactions when they started using gamification. The positive reactions included excitement, curiosity, interest, and challenge with the new system. Since they had never used a gamification system before, the curiosity to know what the new system can bring made them feel excited. The positive reactions are represented in the following quotes:

> *"I'm just excited because I think it's something new. I'm kind of challenged with this new learning site because we don't have a model like this yet." (R1)*

> *"This is something unexpected that it turns out that these gamification models can be applied in learning. And that's good." (R9)*

However, 30% of the participants had neutral reactions, as shown in this quote:

> *"To be honest, at first I think it was just like a usual e-learning site. At that time, when I looked around, it was still like a beta version, so I was still figuring out, I found the maximum points to be achieved were only 5000 points, so I wasn't excited at first." (R3)*

The remaining 50% of the participants expressed their concern and doubt about the new gamification system. Feeling unconfident and confused about using the new system were some negative reactions found, as shown in the following quotes:

> *"I think it will be harder to follow the new system". (R5)*

> *"At first, it was more difficult for me to adjust it because I didn't understand the system yet". (R10)*

> *"I was confused at first because everything was new, there were points, games, and many things to do". (R6)*

Theme 4 centers on the students' behavioral change from negative to positive behavior. There are two sub-themes related to positive behavioral change, as can be seen in Figure 3. As mentioned above, 50% of the participants had negative reactions to the new system. The negative reaction was considered a common reaction when facing changes, as explained in the change curve model [57]. This model states that most people will go through several stages when adjusting to change. The stage begins with the shock or denial

reaction, which will be followed by a fear or anger reaction. If managed well, that stage will continue to the acceptance stage. The acceptance stage could be seen in students' behavior change after using the system for some time. Once the students understood how the gamification system worked and became more familiar with the system, they became more enthusiastic and felt excited. Overall, 70% of all participants that had negative or neutral reactions at the beginning experienced increasing motivation. These increasing motivations can be seen in the following quotes:

> *"At first I was surprised, but after using the system it became more fun because I found new features." (R7)*

> *"Since I already knew how to do this, how to do that, I started to get excited to level up. There were times when I started to want to be on top, just want to beat the others." (R3)*

> *"After doing it and understanding the existing rules, I just feel easier because this will also affect the final score and after doing it, I became more excited." (R5)*

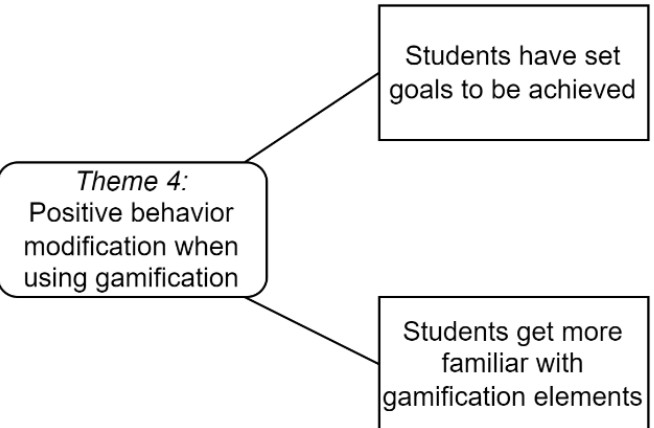

**Figure 3.** Theme 4 and its Sub-Themes.

Another factor that contributed to students' positive behavioral change was students' awareness of individual goals. Students who did not have a clear goal at the beginning began to set the goal for achievement during the learning process, as stated in the following quote:

> *"I started to have a target since the lecturer announced the graduation requirements, and since 3 weeks ago my target was to be in the top 5 on the leaderboard."(R5)*

Despite the positive behavioral change, other students experienced an opposite behavioral change. Theme 5 relates to students' negative behavioral changes. Theme 5 consists of four sub-themes, as can be seen in Figure 4; 30% of the participants who had positive or neutral reactions at the beginning had a declining motivation over time. This finding is consistent with Berkling and Thomas [33], who found a group of students lost their interest in gamification over time as they used the system. Other studies reported similar results, adding a gradual loss of motivation [27,58]. A behavioral modification, particularly from positive to negative behavior (i.e., students who were enthusiastic at first become less enthusiastic), may be triggered by particular circumstances that lead to an unpleasant experience. For instance, when the game rules changed or when the gap between someone's score and the scores of the top 10 students on the leaderboard became broader, as can be seen in the following quotes:

> *"My unpleasant experience was when the rules changed, I had to rearrange my strategy to pass this class." (R4)*

> *"In what week, our lecturer increased the target point. From that time, it felt like it was just like that. I'm not interested in chasing point anymore." (R2)*

*"For me, at first, it felt like there was a sense of competitiveness, but as time went on, that sense faded. Because the gap between my score and the top 10 was getting farther away, it feels like whatever I get is enough. I just do my best." (R1)*

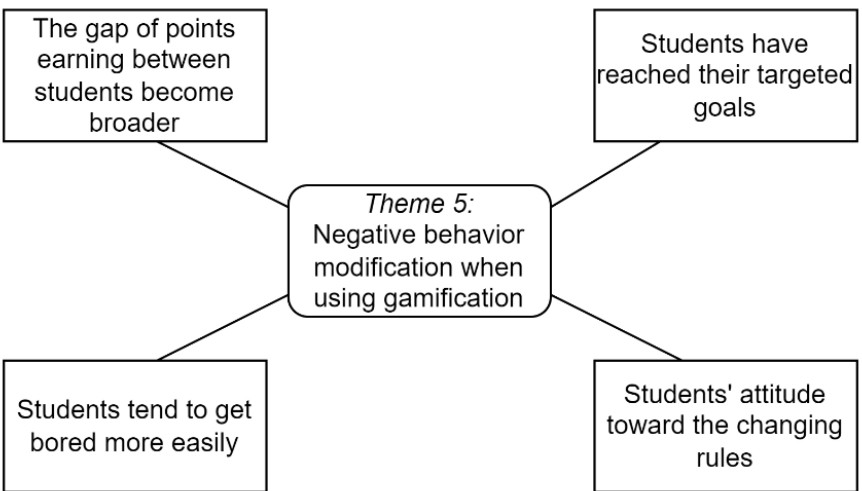

**Figure 4.** Theme 5 and its Sub-Themes.

Besides the unpleasant experience, boredom may also reduce students' motivation. This quote represents that phenomenon:

*"For the first time it is fun, but as time goes by, even though it was cool, I got bored. I don't know why I just got bored." (R4)*

Csikszentmihalyi [59] stated that boredom can be caused by a lack of challenge, or can arise as a result of a combination of high ability and low task demands. In the experiment, 30% of the participants who had high academic performance experienced declining motivation over time. This might be the result of a presence of high ability and low task demands that they experienced when using the gamification system. Some activities to collect points (e.g., updating the wikis, or displaying the materials) were considered activities that lacked challenge, particularly for students with high academic performance abilities. Fisher [36] stated that monotonous and repetitive activities that lack complexity, variety, and cognitive stimulation may be the cause of boredom. The use of farming activities, such as collecting points by displaying the materials repetitively, may lead to a boring experience. The repetitive nature of farming activities was considered monotonous for 20% of the participants, as shown in the following quote:

*"After reaching a certain point, it feels like it's just like that, and the things we do to get points are monotonous. So at first I'm happy but as time goes, I've just wanted to finish the task." (R2)*

For students who had targeted goals, for example, to achieve some particular points, when they had reached their targeted goals, their motivation declined. Students felt that after they had reached their target, there was no need to pursue the points anymore, and their participation in the class decreased. This quote represents that phenomenon:

*"Because I have reached my target, I'm not interested to do anything else. Even though the maximum level is still far away, for me it's over." (R4)*

Theme 3 centers on students' preferences for an e-learning system. Despite various behavior changes that occurred during gamification implementation, all students agreed that they preferred the gamification system to conventional learning, as represented in the following quotes:

*"I prefer gamification, because it's a new experience to me, and because there is a point system and games like that. So we don't just upload assignments, we have other tasks to do, like doing games and looking for points." (R5)*

> *"I prefer gamification. Because for me, it will be more exciting if there is a reward for something we do." (R1)*

> *"I prefer this new LMS to the conventional one. It's more exciting like this. The new system is more motivating and engaging than the old one. It's proven by the number of students that are more likely to access the new system than the old one, which we often use only to upload assignments." (R3)*

However, there is an exception for selecting learning with gamification, as opposed to conventional learning, as shown in this quote:

> *"If I have to choose, I may choose gamification. But if there are several courses, for example, 5 courses that use gamification simultaneously in one semester, it's a bit of a hassle in my opinion, because we will be busy chasing points in all courses." (R9)*

From this quote, we can infer that although students preferred learning with gamification to conventional learning, they could see the drawback of taking some gamification classes simultaneously. Students felt that they needed to exert more effort to carry out gamification activities compared to conventional learning, so if they took several classes with gamification, it could become a burden for the students.

### 4.2. Students' Perception of Gamification Elements and How It Influences Motivation and Engagement

In this sub-section, we will present theme 2, which is related to the students' perception of gamification elements and they influence motivation and engagement. Theme 2 consists of five sub-themes, as shown in Figure 5. From the analysis, we found the four most important game elements for the students. Those elements are points, leaderboards, badges, and gamified tests. Points are a representation of the general reward feature. Badges represent the meta-game reward. The leaderboards are used to accommodate students' social needs, whereas gamified tests empowered practice implementation that accommodates students' competency needs. The findings revealed that 50% of the participants considered gamified tests as the most important element, 20% of the participants considered points as the most important element, 20% of the participants considered the leaderboard as the most important element, and the remaining 10% considered badges as the most important element of gamification.

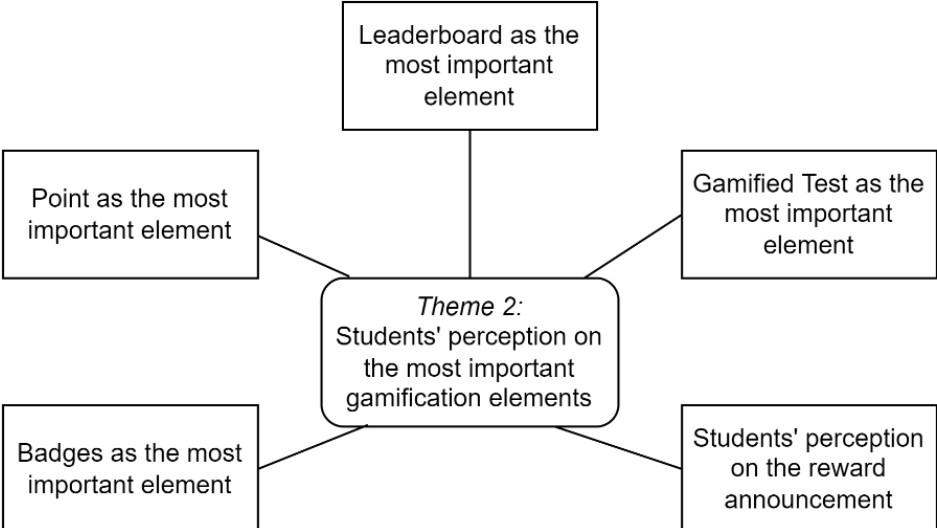

**Figure 5.** Theme 2 and its Sub-Themes.

### 4.2.1. Points

A point is a general reward that is given to students after they have carried out a particular activity. Points play an important role since they can determine leaderboard position, badge earning, and graduation from the course. The minimal requirement for

students to graduate from the course was to reach level 5 or gain 3500 points. Students expressed various feelings when they achieved a point; 30% of the participants who felt excited at first experienced a gradually fading feeling after they had achieved the minimal target point.

> *"At first, I was still excited, I think "Wow, this is fun," But just the more you get used to it, it seems like it is just a usual thing." (R6)*

> *"Before I reach the required minimum point, I feel happy. But after reaching the minimum point, it feels like yes, it's good if you get it, but if not, it is just fine." (R2)*

The excitement of getting a point at the beginning may be the result of the novelty effect of the feature. As students became used to gaining points, it became something usual. This behavioral phenomenon is called habituation [60]. Habituation refers to the decline in response to a feature when that feature is no longer novel and has no rewarding consequences [61]. The reward system wherein students will gain particular points for every activity made students feel that getting points was no longer an issue of pride for them. However, not all students showed decreasing excitement when collecting points; 40% of the participants who did not feel enthusiasm at the beginning showed excitement when collecting points, as said in the following quote:

> *"I started chasing points from the mid-semester. I was looking for our lecturer's attention. If there was a comment or post on the Forum, I would reply right away. I'm like being more enthusiastic." (R3)*

A review study on gamification [62] stated that the points and leaderboard, common gamification elements that have been criticized for only providing extrinsic motivation, are capable of fostering enthusiasm, a sense of enjoyment, fun, and general positive feelings towards learning, values that are directly associated with intrinsic motivation.

Our data have revealed that students who maintain their performance during learning with gamification are students that can see "the value of the point", not just the point itself. It was not the point that made them excited, but rather the privilege that they might derive by having many points. For example, students who successfully managed to be the first three to get level 10 (20,000 points) were awarded the Top Scorer badge. Students who were awarded the Top Scorer badge achieved the minimal course grade "A-". This quote represents the phenomenon:

> *"I keep pushing to get more points to get the privileges. I want to have a good grade in this course and I also want my team to be in the first rank." (R9)*

Other students valued points as proof of their hard work and as a trigger to increase their enthusiasm, responsibility, and participation in learning, as described in these quotes:

> *"If there are no points, I rarely open the e-learning site. So points do give a little bit of enthusiasm." (R8)*

> *"I think it's good to know how many points our friends have. So we can increase our enthusiasm to beat our classmates by getting more points and increase our level. We become more active in the class." (R9)*

> *"We feel like we have a responsibility to increase our points every day, so we don't lag behind the others." (R10)*

> *"Point is the most important feature in my opinion because points can be the proof of our effort." (R6)*

The gamification system employed many mechanisms for collecting points. One of them was farming practice. Farming is a repetitive activity that students can undertake to achieve some points. In this system, farming is represented by some activities, including posting wiki, displaying the learning materials, and downloading the learning materials. The goal of implementing farming practice in the gamification system was to enforce students' behavioral engagement. The farming practice was designed so that the students could access the e-learning site more frequently. This goal was successfully achieved. From

the e-learning log, we can see that the total e-learning access for one semester was 140,614. This number is much higher than the total e-learning access for another class that did not employ gamification, which was only 11,034 for the same semester. In total, 40% of the participants were so engaged with collecting points that they undertook the farming activities every day, even until late at night, as can be seen in the following quotes:

*"I can be online for hours. I even make a timer so that every hour I can remember to fill out the wiki, post a message, or display the material to get some points." (R3)*

*"I always maximize the maximum limit of points that can be obtained every hour, so I will go online almost every hour to look for points, even staying up late at night." (R3)*

*"When I was very excited to chase points, I almost online 24 h every day. Sometimes I asked my friend to login to the system for me when I can't." (R7)*

Even though the farming practice may develop students' behavioral engagement, in the learning context, the behavior may harm the main goal of the learning itself. Students' attention may shift from learning to collecting points, as represented in this quote:

*"For me, the most important thing about this course is to get as many points as possible, the other thing isn't that important anymore." (R3)*

While the activities of collecting points have been set with activities related to learning, such as posting a wiki and displaying the materials, the students tended to do this just to get some points, and not to try learning the materials. This finding is similar to that from a study by Baydas and Cicek [63], revealing that students were more focused on earning badges, reaching a high rank on the leaderboard, or just being successful, rather than learning the content during the gamification process. From the current study's findings, we can see that the use of points in gamification could foster students' motivation and behavioral engagement. However, there is a drawback to using points to stimulate students' motivation. Future gamification using points must be designed carefully to minimize the negative effect so that students can stay focused on their learning.

### 4.2.2. Leaderboard

The leaderboard is the most cited gamification element within the data. This finding is consistent with a previous review study's findings showing that the leaderboard was the most cited game design element among 11 other elements [64]. The leaderboard was the must-visit page when students opened the e-learning site. The behavior of regularly visiting the e-learning site to check the leaderboard led to students' behavioral engagement. Leaderboards trigger a student's competitive spirit, as said in the following quotes:

*"It's good that there is a leaderboard like that, so we can compete with each other in the class." (R10)*

*"Every time I see a change in the leaderboard, I feel like, how come this could be higher, they have more points than me, so it's like I'm more motivated to chase points." (R1)*

The leaderboard acting as a driver for a competitive environment was also mentioned in other studies [65,66]. It serves as a feedback mechanism for social competition, and may promote the engagement of participants [67]. In a competitive environment, students tend to compare themselves with other students. Social comparison theory [68] may explain the social comparison induced by leaderboards. According to the theory, people continually compare themselves with others, as this is a fundamental psychological mechanism that affects people's judgments and behavior [66]. Students can maintain or enhance positive self-views by comparing themselves with inferior others (downward comparison) [69]. Otherwise, students can also obtain information about their relative standing and how to improve themselves by comparing with superior others (upward comparison). The upward comparison can motivate people to work better [9], but it may also threaten individuals' self-views [70]. Seeing themselves in a lower position may decrease students' motivation, as said by R4:

*"At first, I had a target to get a position 1–5, but after some times the position was overtaken by others, so I just didn't want to pursue it anymore."*

For students who have usually been in the top position of the ladder, seeing the leaderboard offers a reminder to maintain their position, as said by R3:

*"When we see the leaderboard, we often feel anxious, because we are worried whether our position has been passed by other team or not. So we are as a team always reminding each other to maintain our position."*

We can see that leaderboards may foster students' motivation by raising the spirit of competition. By knowing their position on the leaderboard, students were motivated to raise or maintain their position. However, this phenomenon was not experienced by a group of students who easily felt discouraged. From the study, we also found that goal setting may play a role in students' motivation in achieving the top level on the leaderboard. The goal-setting theory explains that goals are effective if people are committed to them, and performance is maximized when individuals are committed to difficult, specific goals [71]. In the current study, students who held the top position on the ladder set their goals for achievement. For instance, a student had a goal to within the top three positions of the ladder, while others set their goal as being in the top five of the ladder. This goal encourages them to work hard to reach their desired position. This finding is consistent with a study by Landers et al. [72] that found that the leaderboard made the relationship between goal commitment and performance stronger.

### 4.2.3. Badges

Each student had a different perception about what badge was most important to them. In total, 50% of the participants considered the team badge as the most important badge, 30% of the participants argued that the champion badge was the most important, and 20% of the participants chose the top scorer badge as their most important badge. This difference in perception may have been caused by the different values that students expected to obtain by owning a particular badge. For instance, a student felt that getting a team champion badge was the most important since it could boost a student's sense of pride by becoming a member of the best team in the class. Other students considered the champion and the top scorer badges as the most important badges since they could prove their achievement to their classmates. These quotes represent the phenomenon:

*"The team champion is the most important because it is the means to prove that our team works harder than other teams." (R2)*

*"For me, the champion badge which needs 20,000 points is the most important badge." (R7)*

*"The most important badge is the top scorer badge. That badge is important because we can boast ourselves as the first five students who managed to get the top score." (R3)*

Despite the different perceptions about what badge is the most important, all students agreed that badges play a role as proof of students' achievement. The positive effect of badges in increasing students' motivation and engagement has long been studied [73,74]. In the current study, badges could boost students' pride, and as a consequence, they tend to be more enthusiastic in carrying out the course, as can be seen in the following quotes:

*"Receiving a badge makes me proud and makes me more enthusiastic in studying this course." (R9)*

The data reveal that it is not the badges themselves, but the "value" of the badges, that motivates students to work hard in the course, as said by R3:

*"Getting a badge is not the important thing to me. Just because I see it is useful, for example, you will get an "A", if you get this badge, and you will get a "B", if you get that badge, I become more interested".*

In our experiment, we set some privileges for students who achieved particular badges. For instance, students who achieved the "Champion" badge will automatically get an "A", and students who get the "Team champion" badge will get an "A-" for the course. This

finding reveals that students were encouraged to get a particular badge to get the desired reward. Related to its capability to boost students' pride, the students' feeling of pride was reduced if their earning of the badge was not announced in the class. In total, 30% of the participants felt more proud when their classmates acknowledge their achievements, as represented in the following quotes:

*"Badges become unimportant if not announced because what's the point of getting a badge if your friends can't see it." (R3)*

*"If it is announced, it makes me feel happy. This badge is proof that I have achieved the target. My friends will see and appreciate it." (R5)*

Announcing student badge-earning would satisfy students' need for appreciation. Fagley [75] argued that appreciation might play an important role in mental health and affective well-being. Appreciation is also viewed as related to spirituality and as a significant ingredient for success in the workplace. However, individuals show differences in their tendency to feel appreciation [76,77]. One may feel that getting a congratulation from his/her classmates is a big sign of appreciation, whereas others may not feel that way, as said by R7:

*"For me, whether to announce it or not, it doesn't matter. The important thing is that the badge can be exchanged for grades."*

It is not the value of appreciation from the classmates, but the value of getting a good grade from earning the badge, that is considered important to R7. Despite the various values the badges hold, the badges successfully attract and motivate students to get them.

### 4.2.4. Gamified Test

The last gamification element that is deemed important to students is gamified tests. According to a review study by Kalogiannakis et al. [62], most studies reviewed focused on gamification elements such as competition, leaderboard, points, and badges, leaving other elements such as quizzes, with limited exposure to their potential. In the present study, the quizzes were presented in gamified forms. Gamified tests that came in various forms every week made students more enthusiastic and motivated to undertake the test, as R2 said:

*"I become more motivated to do the test. Gamified test is the one I look for every week."*

The goal of the gamified test is to test the students' understanding of the learning materials. It is expected to satisfy students' need for competence. Competence is one of the three basic psychological needs—autonomy, competence, and relatedness—of Self-Determination Theory (SDT) that are essential for well-being and psychological growth [78]. Since it is directly related to competency, most students agreed that the gamified test was important, as stated in R9's quote:

*"I think the most important feature is the gamified test because it can trigger us to learn the material. This should be the main goal of gamification, which is to make students more enthusiastic about learning the materials."*

The unlimited attempts possible for the gamified tests allowed students to study the materials repeatedly so they could better understand the materials, as R2 said:

*"We have to try and sometimes repeat several times to find the right answer. But it has a good impact, we can memorize the material being tested better."*

These findings imply that the gamified test is an effective means to increase students' motivation. The gamified test provides freedom for students to undertake it at any time and however many times they want. This mechanism may satisfy students' need for autonomy. The satisfaction of three basic psychological needs, particularly autonomy and competence needs, plays a key role in enhancing intrinsic motivation [79]. The existence of gamified tests also successfully fosters students' cognitive engagement. Cognitive engagement relates to how much effort students apply to master certain ideas or skills [80]. Students were indirectly pushed to study the materials by undertaking gamified tests every week, as said by R1:

*"I think it's innovative and pushes students to read the material. Because if you want to play the game, you have to read the material first. Then you can play the game."*

The burden of learning can be reduced by undertaking the tests through games, which made students feel less challenged and more excited to learn, as said by R9:

*"Doing the test through games makes the learning more easy and fun."*

The quote reveals that gamified tests have successfully changed learning into an enjoyable experience. Thus, the finding implies that one goal of gamification, to make learning more fun, has been achieved.

*4.3. The Influence of Population Characteristics on Motivation and Engagement*

The present study employed a gamification system intending to increase students' motivation and engagement. However, the effectiveness of the gamification applied may be influenced by contextual factors. One of the factors is the population's common characteristics. Theme 6, which consists of three sub-themes, centers on the common characteristics of final-year students (Figure 6). The first characteristic is that final-year students tend to focus on a goal oriented toward their study completion, or focus on getting skills that will be needed after they graduate. This characteristic made them more interesting in a subject they deemed more important for their future. For instance, for students who took the course (with gamification) along with the internship, performing gamification activities was less important than doing the internship tasks. As a result, they were not very motivated to undertake the gamification activities. This can be seen from the following quote:

*"I don't care about earning points, because at the same time I'm also doing my internship, so I'm focused on working on my projects." (R2)*

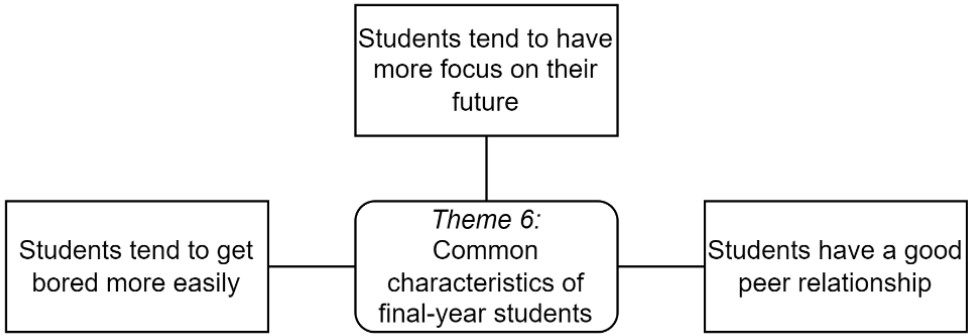

**Figure 6.** Theme 6 and its Sub-Themes.

For students such as R2, gamification did not effectively motivate them since they considered having more important things to do. The challenge for future gamification design for final-year students involves how to make students consider gamification activities as equally important as other activities undertaken to achieve their future goals. The gamification design must also pay attention to how much work students must do in the gamification system. Some gamification activities, i.e., farming activities that require students to perform repetitious activities, may demand a lot of students' time and effort. Those activities may also cause students to feel burdened. The selection of gamification activities in the future, particularly for final-year students, must consider how much the activities will add to the burden placed on the students.

The second characteristic of the population is that the final-year students had a better relationship with classmates and lecturers than their juniors. Since students had spent a long time together, they had developed an established friendship group. It thus became easier to work with classmates since they already knew their classmates well. The existence of teams and team-based activities makes the relationship between students stronger. The use of gamification elements that involve teamwork has increased students' social and emotional engagement. Students tend to help and support their teammates to achieve more

points or higher ranks. A member of the team can help one another in the learning process, as said by R14:

> *"It's more exciting and more fun if you have a team. If you're alone, it's like empty. We able to share and help each other as a team. I was helped a lot with my teammates and friends outside the team."*

With a good relationship that has been built before, students tend to work well together. The existence of team activities may increase students' performance. This phenomenon implies that gamification design for final-year students may highlight various team-related activities to boost students' performance.

The third characteristic of the population is that most of the final-year students tend to have less enthusiasm when attending a course compared to their juniors. They also get bored more easily compared with the lower-level students. Since they have been studying for a long time, attending a course becomes a monotonous and repetitious activity that they must take to complete their study. The challenge for future gamification design for final-year students is how gamification can create variable enjoyable experiences for students [81]. Introducing new activities, implementing variable rewards, and increasing the value of the reward, are some of the things that can be done to maintain students' enthusiasm. Variable rewards are one of the phases of Eyal's Hook Model that aim to maintain user interest by sustaining variability in giving rewards [82]. The variability can be realized in the type of rewards or the reward schedule. Increasing the value of the reward means giving values to the reward that are deemed more important for final-year students. For example, if a student gets the Top Scorer badge, they will get additional assistance from the lecturer in working on their thesis.

However, even after gamification implementation, 30% of the participants in our research still failed to maintain their enthusiasm for study. This particular group of students that are unmotivated to learn might perceive gamification as a motivation to learn. According to causality orientation theory [83], a sub-theory of SDT, people differ in the extent to which they experience their actions as self-determined, which further influences whether they perceive feedback as informational or controlling. Hence, a person's causality orientation may influence the effects of feedback on need satisfaction. Autonomy-oriented individuals interpret external events as informational rather than controlling, therefore experiencing more competence need satisfaction. Control-oriented individuals, in contrast, perceive external events as pressuring, and therefore experience fewer feelings of autonomy. A person's causality orientation may further moderate how feedback affects need satisfaction and intrinsic motivation [84]. Students with high control orientation may perceive gamification as controlling, and therefore have decreased intrinsic motivation [85].

## 5. Conclusions

This current study employed a qualitative methodology to (1) examine the students' behavioral change when using e-learning with gamification, (2) investigate gamification elements that are important to students and how they influence students' motivation and engagement, and (3) investigate whether population characteristics may influence students' motivation and engagement. The findings revealed two distinct behavioral changes in students when using the gamification system, namely, changes from negative to positive, and changes from positive to negative behavior. Despite the distinction of behavioral changes that occur during gamification implementation, all students agreed that they preferred the gamification system to conventional learning. The findings also reveal four gamification elements that were deemed important for students, namely, points, leaderboards, badges, and gamified tests. Those elements are considered important for students since each element can bring value to students. An awareness of the values of the elements increased the students' motivation. The attempt to collect points, such as displaying the materials or updating wikis, which can be done repeatedly, may lead to students' behavioral engagement. The behavior of regularly visiting the e-learning site to check the leaderboard leads to students' behavioral engagement. The use of the

leaderboard fostered students' motivation by raising the spirit of competition. Badges motivated students by boosting their pride. Similar to points, the findings also reveal that gamified tests could satisfy students' need for competency and autonomy, thus enhancing students' intrinsic motivation. The existence of gamified tests that pushed students to learn the materials regularly may foster students' cognitive engagement. We also found that the use of gamification elements that involve teamwork has developed students' social and emotional engagement.

Despite the finding that gamification has enhanced students' motivation and engagement, we also found two main concerns about using the gamification elements that may impact students' performance and motivation. The first concern is related to the use of points and farming activities. The points and farming activities may harm the learning goal by making students' focus shift from learning to collecting points. Farming activities that force students to do repetitive actions regularly may lead to a boring experience that may further impact students' motivation. Farming activities also demand a lot of the students' time, which may impact students' motivation, particularly for students who do not have much time to carry out the gamification activities. The second concern relates to the factors that may lead to decreasing motivation. Students may only be enthusiastic about using the system because of its novelty effect, so it may be difficult to maintain their motivation over time. The decreasing motivation may also be caused by the changing rules, the gap in the points earned or the gap in the leaderboard position, the attitudes of students who easily get bored, and the students' assumptions that they have achieved their goals.

We also found population characteristics, i.e., final-year students have a great influence on gamification effectiveness in relation to its capability to increase motivation and engagement. The final-year students tended to focus on activities that are considered more important for their future. This fact implies that the design of gamification activities for final-year students must consider how to make the gamification activities as important as other activities that students deem important to achieving their future goals. The other characteristic of final-year students is they have had a better relationship with their classmates and their lecturers. The consequence of this fact is that the use of gamification elements that involve teamwork is more effective for motivating and engaging students. The last characteristic of the final-year students is that they tend to have less enthusiasm for learning or attending a class. Creating variable enjoyable experiences may be pursued to maintain students' enthusiasm.

From our study, we can suggest that future gamification design must consider using gamification elements that can sustain students' motivation longer by dealing with the factors that may cause students to lose their enthusiasm. The use of gamification elements such as points and farming must be implemented with caution so that students can remain focused on learning, not just collecting points. Future gamification must also consider whether the use of farming activities will bring benefits or not, since it may cause boredom and take a lot of the students' time. The design of future gamification classes must also consider the population characteristics so that the design of gamification elements may foster the effectiveness of gamification to enhance motivation and engagement.

**Author Contributions:** Conceptualization, F.S.R., L.E.N., R.F. and D.B.S.; methodology, F.S.R., L.E.N., R.F. and D.B.S.; validation, F.S.R., L.E.N., R.F. and D.B.S.; formal analysis, F.S.R.; investigation, F.S.R.; resources, F.S.R.; data curation, F.S.R.; writing—original draft preparation, F.S.R.; writing—review and editing, F.S.R., L.E.N., R.F. and D.B.S.; visualization, F.S.R.; supervision, L.E.N., R.F. and D.B.S. All authors have read and agreed to the published version of the manuscript.

**Funding:** This research received no external funding.

**Institutional Review Board Statement:** The study was conducted in accordance with the Declaration of Helsinki, and approved by the Ethics Committee of Universitas 'Aisyiyah Yogyakarta (No. 2150/KEP-UNISA/VI/2022).

**Informed Consent Statement:** Informed consent was obtained from all subjects involved in the study.

**Data Availability Statement:** Not applicable.

**Conflicts of Interest:** The authors declare no conflict of interest.

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
