# Peer review of "Motivation and Engagement of Final-Year Students When Using E-learning: A Qualitative Study of Gamification in Pandemic Situation"

_sustainability, doi:10.3390/su14148906_

Round 1
Reviewer 1 Report
The article presents a study on the impact of a gamification strategy on student motivation and engagement in a pandemic situation and online teaching. It is a relevant, current topic.
1. introduction
The introduction is complete, relevant and focused on the research problem. The careful study of the effects of gamification should be highlighted, not forgetting the studies that have shown less positive results with its use.
2. Materials and methods
The description is adequate.
3. Gamification Implementation in E-Learning
The description of the implementation is sufficient, although it lacks something that frames and relates all the gamification strategies that are later referred to in the text. It would be very interesting to have a global map/diagram of the gamification experience, including all the components with gamification and the related metrics.
4. Results and Discussion
The results presented are interesting, however, even though this is a qualitative study, there is a lack of quantification to enable the dimension of the negative and positive impacts to be understood in the different aspects under analysis. As most of the results are presented, it is difficult to understand whether the overall impact can be considered positive or negative, as we do not have a clear view of the size of the groups that expressed a positive or negative opinion of the introduction of the gamification strategy.
It is suggested that the article could quantify the dimension of the groups of students with a positive or negative trend position for each dimension under analysis. It should avoid expressions like "some students".
5. Conclusions
After reading the results presented, the conclusions do not seem to take into consideration most appropriately the negative positions that were reported. Perhaps this perception of mine is related to the aspects mentioned in the previous point, regarding the lack of a view of the dimension of the groups with positive and negative positions about the various dimensions under analysis.
It would be appropriate to have more balanced conclusions, including some reservations regarding some of the strategies that were adopted, which deserved more critical reflections from the students. This also should be taken into consideration for the abstract.
Reviewer 2 Report
Name of the paper:
Motivation and Engagement of Final-Year Students when Using e-Learning: A Qualitative Study of Gamification in Pandemic Situation
This paper examines the gamification elements and students' behavioral change when using e-learning with gamification. It investigates gamification elements and claims to influence students’ behavior, motivation, and engagement. They collect data using a qualitative method. Using thematic analysis, they found significant impacts on students’ motivation, behavioral engagement, cognitive engagement, and social-emotional engagement.
.General Observation:
a) There are some typos and grammatical errors in the manuscript, for instance. Research Methodology may be “research methodology," please check Line 219 spacing….
b) “The present study involved 22 final-year students who attended a research methodology class in the Information Systems Program”. The sample size looks very small to claim the results.
c) Please refer to :”…game playing [35], [36], [37], [38], [39]. “ Could be “…..game playing [35-39].”
d) Please refer to Table 1: Sub-features may be confused with element types hence they may be renamed as sub-elements.
e) Please refer to: "We used the taxonomy... manipulation and control features." Some terms need explanation or clarity for further understanding.
f) Please refer to 690” “Our previous study has implemented a gamification system in a class that was held..." This does not give any clarity whether the authors are referring to published or unpublished work, or work in this research?
g) Please refer to the reference: “Braun, V., & Clarke, V. (2021). One size fits all? What counts as quality practice in (reflexive) thematic analysis? Qualitative research in psychology, 18 (3), 328-352. Instead of reference no. [34] as Braun, V., & Clarke, V. changed the phases of reflexive thematic analysis.
h) The authors may provide all stages considered in the thematic analysis using Nvivo. The authors may provide the analysis using the phases of Nvivo (as they claimed) validated.
Reviewer 3 Report
Dear authors,
Your work is well-prepared, and the study is interesting and offers important findings on the topic. Below, I make some suggestions for improvement. Well done!
Specific Comments:
Line 40-61: Please enrich your references with more studies concerning the students’ view on the pandemic educational settings. Interesting papers you can find in the Special Issue "The Role of Technology in Teaching, Learning, and Assessment during and Post-COVID-19: Opportunities for Innovation and Challenges" Education Sciences (https://www.mdpi.com/journal/education/special_issues/Role_Technology_Teaching_Learning_Assessment)
Results and Discussion: discuss also with Kalogiannakis, M.; Papadakis, S.; Zourmpakis, A.-I. Gamification in Science Education. A Systematic Review of the Literature. Educ. Sci. 2021, 11, 22.
Line 690: Reference for your previous study
Conclusion: The conclusions highlight the positive findings but I think that it is important to mention some of the students’ concerns (f.e. novelty effect, the lack of motivation that points could create and so on). I suggest to make your conclusion more balanced.
General comment:
See again issues with typos and fonts (f.e. abstract, a lot of spaces that you don’t need and so on).
Reviewer 4 Report
This manuscript was easy to read and interesting. The authors present a well-organized, clearly-written Introduction. Though not critical, sub-headings may help, though the transitions are pretty obvious. This reviewer suggests on lines 68-70 checking the numbers (21 studies, 20 positive findings, 12 negative); they might be correct, just verifying.
Under Methods, this reviewer would like a little more information on how and why students were selected for the second focus group. Also, was the qualitative analysis done by just one researcher, were there any checks by other researchers for agreement? It is a little unclear; line 174 just says "We then re-read".
Under Gamification, think lines 211-221 are repetitive. There is a misspelling dan -> and on line 229.
The Results and Discussion section is quite comprehensive.
The Conclusions are fine. This reviewer would like to see an Implications section or subsection with ideas of what was learned and how to apply those findings to a future class.
Overall well done.
Round 2
Reviewer 2 Report
The authors have responded well to all queries I raised except quarry no.2 about small sample size, which looks to me the only negative point for this paper. Thank you Dr M N Qureshi
